# Pro-metastatic collagen lysyl hydroxylase dimer assemblies stabilized by $Fe^{2+}$-binding

Hou-Fu Guo [1], Chi-Lin Tsai [2], Masahiko Terajima [3], Xiaochao Tan[1], Priyam Banerjee[1], Mitchell D. Miller [4], Xin Liu[1], Jiang Yu[1], Jovita Byemerwa[1], Sarah Alvarado[4], Tamer S. Kaoud[5,6,7], Kevin N. Dalby[5,6], Neus Bota-Rabassedas[1], Yulong Chen[1], Mitsuo Yamauchi[3], John A. Tainer [2], George N. Phillips, Jr.[4] & Jonathan M. Kurie[1]

Collagen lysyl hydroxylases (LH1-3) are $Fe^{2+}$- and 2-oxoglutarate (2-OG)-dependent oxygenases that maintain extracellular matrix homeostasis. High LH2 levels cause stable collagen cross-link accumulations that promote fibrosis and cancer progression. However, developing LH antagonists will require structural insights. Here, we report a 2 Å crystal structure and X-ray scattering on dimer assemblies for the LH domain of L230 in *Acanthamoeba polyphaga mimivirus*. Loop residues in the double-stranded β-helix core generate a tail-to-tail dimer. A stabilizing hydrophobic leucine locks into an aromatic tyrosine-pocket on the opposite subunit. An active site triad coordinates $Fe^{2+}$. The two active sites flank a deep surface cleft that suggest dimerization creates a collagen-binding site. Loss of $Fe^{2+}$-binding disrupts the dimer. Dimer disruption and charge reversal in the cleft increase $K_m$ and reduce LH activity. Ectopic L230 expression in tumors promotes collagen cross-linking and metastasis. These insights suggest inhibitor targets for fibrosis and cancer.

[1] Department of Thoracic/Head and Neck Medical Oncology, The University of Texas MD Anderson Cancer Center, Houston, TX 77030, USA. [2] Department of Molecular and Cellular Oncology, The University of Texas MD Anderson Cancer Center, Houston, TX 77030, USA. [3] North Carolina Oral Health Institute, School of Dentistry, University of North Carolina at Chapel Hill, Chapel Hill, NC 27599, USA. [4] Department of Biosciences, Rice University, Houston, TX 77251, USA. [5] Division of Medicinal Chemistry, Targeted Therapeutic Drug Discovery and Development Program, College of Pharmacy, The University of Texas at Austin, Austin, TX 78712, USA. [6] Division of Chemical Biology & Medicinal Chemistry, College of Pharmacy, The University of Texas at Austin, Austin, TX 78712, USA. [7] Department of Medicinal Chemistry, Faculty of Pharmacy, Minia University, El-Minia, 61519, Egypt. Correspondence and requests for materials should be addressed to J.A.T. (email: jtainer@mdanderson.org) or to G.N.P.Jr. (email: georgep@rice.edu) or to J.M.K. (email: jkurie@mdanderson.org)

Work in pancreatic cancer models has shown that intratumoral fibrosis exerts mechanical forces that activate tumor cell invasive properties and restrict the influx of immune cells and vascular cells, creating an immuno-suppressive, hypoxic environment that favors metastasis and inhibits antitumor drug delivery[1–4]. The intratumoral fibrotic reaction results from a proliferation of fibroblasts that differentiate into myofibroblasts and produce a collagenous matrix that stiffens the tumor stroma owing to covalent collagen cross-links that amass over time[1,5,6]. These findings led to the working hypothesis that metastasis results from an accumulation of collagen cross-links that stiffen the tumor stroma[1]. However, findings from sarcoma and breast cancer models showed that lysyl hydroxylase 2 (LH2), which hydroxylates lysine (Lys) residues on collagen's N- and C-terminal telopeptidyl domains prior to the initiation of cross-link formation by lysyl oxidase (LOX), promotes metastasis, is a target of hypoxia-inducible factor 1, and contributes to hypoxia-induced tumor stiffening[7–9]. Subsequent findings by our group showed that metastatic lung adenocarcinomas produce a stable type of collagen cross-link driven by high expression of LH2; LH2 depletion in tumor cells led to a reduction in tumor stiffness and metastatic activity and a change in the type, but not the total amount, of collagen cross-links as manifested by a decrease in hydroxylysine (Hyl) aldehyde-derived collagen cross-links (HLCCs) and a reciprocal increase in Lys aldehyde-derived collagen cross-links (LCCs)[10]. Work in connective tissues has shown that HLCCs are more stable than LCCs[11]. In addition to providing a rationale to target LH2 in cancer, these findings support a paradigm in which LH2 functions as a regulatory switch that controls the relative abundance of distinct types of collagen cross-links that influence the metastatic fate of tumor cells.

The collagen LHs include three vertebrate (LH1-3) and one invertebrate gene that are members of a large family of $Fe^{2+}$- and 2-oxoglutarate (2-OG)-dependent oxygenases. These oxygenases utilize mononuclear non-heme iron and the co-substrates 2-OG and oxygen to oxidize substrate and generate succinate and $CO_2$[12]. They are widely distributed throughout the animal kingdom (over 60 in humans alone) and catalyze a variety of oxidative reactions that drive diverse biological processes such as collagen post-translational modification (prolyl hydroxylases and LHs), fatty acid metabolism (trimethyl lysine dioxygenase), oxygen sensing (factor-inhibiting HIF and prolyl hydroxylase domain containing protein 2), DNA and RNA repair (ten–eleven translocation proteins), and demethylation for epigenetic regulation (Jumonji proteins)[13]. Crystallographic studies of numerous family members have revealed conserved features in the active site, where $Fe^{2+}$ is ligated by a 2-His-1-carboxylate facial triad located within a double-stranded β-helix (DSBH) fold, as well as many structural differences. For example, some family members function as monomers, whereas others function as dimers[14]. Findings from crystallographic studies of prolyl hydroxylases and Jumonji family members have led to the development of selective antagonists that are under clinical development. However, collagen LHs have less than 15% homology to other $Fe^{2+}$- and 2-OG-dependent oxygenases, and their structures have not been reported, which represents a major hindrance to LH2 antagonist development.

Here, we sought to gain insight into the structural properties of a collagen LH. We used a collagen LH that was recently discovered at the C terminus of the L230 protein in the giant virus *Acanthamoeba polyphaga mimivirus* (APMV), a human lung pathogen[15]. L230 encodes an enzyme that has distinct LH and glycosyltransferase domains similar to the bi-functional architecture of human LHs and is capable of hydroxylating Lys residues and glycosylating the resulting Hyl residues on collagen[15].

Whether L230 has telopeptidyl LH activity and generates HLCCs similarly to LH2 remains unclear. Our findings reveal features of the L230 active site and dimerization interface that are conserved in human LHs, show that L230 dimer assemblies are stabilized by $Fe^{2+}$-binding, and demonstrate that L230 has telopeptidyl LH activity and promotes cancer metastasis. These findings suggest that LHs have evolved a dimerization mode that adapts to the unique structural properties of collagen.

## Results

**L230 active site and dimer interface are conserved in LHs.** Although its sequence resembles invertebrate LHs (Supplementary Fig. 1a), L230 has >35% homology to vertebrate LHs (Supplementary Fig. 1b and Supplementary Table 1) with similar predicted secondary structure and solvent-exposed regions (Supplementary Fig. 1c). We solved a 2 Å crystal structure of the L230 LH domain (amino acids 680–895), revealing a DSBH fold (Fig. 1a and Supplementary Fig. 2) resembling those of other $Fe^{2+}$- and 2-OG-dependent oxygenases[16]. However, DALI structural search identified modest homology to other oxygenases; the top hit was a putative oxygenase from *Shewanella baltica* with 2.2 Å rmsd and 14% sequence identity (PDB ID: 3DKQ). By comparison to other oxygenases, L230 has unique insertions within the DSBH core and the N- and C-termini.

$Fe^{2+}$ is coordinated by an active site amino acid triad (His825, Asp827 and His877) (Fig. 1a). Tyr814 and Arg887 are positioned to bind the 2-OG C-5 carboxyl (Fig. 1a, inset). These residues are largely conserved among $Fe^{2+}$- and 2-OG-dependent oxygenases[16]. Other side chains facing the 2-OG-binding pocket are conserved in human LH1-3 but not invertebrate LHs (Supplementary Fig. 3 and Supplementary Table 2), except L230 Ala879, which is glycine in human LHs (Fig. 1b, c). An enzymatic activity assay employing bovine skin collagen substrate to measure L230-induced conversion of 2-OG to succinate showed that activity was extinguished by mutation of $Fe^{2+}$- and 2-OG-binding residues but not non-conserved Ala879 (Fig. 1d).

The L230 LH domain formed a tail-to-tail homodimer (889 Å² interface) through loops in the DSBH core (Fig. 2a). In solution, L230 LH domain formed dimers based upon small-angle X-ray scattering (SAXS) experiments (Fig. 2b, Supplementary Figs. 4 and 5, and Supplementary Table 3). The $K_d$ value for L230 LH domain was $23.9 \pm 2.4$ nM by microscale thermophoresis analysis (Supplementary Fig. 6). Importantly, this shared dimerization mode is different from other dimeric β-helix core-containing oxygenases[17–19]. Factor-inhibiting HIF and the ribosomal oxygenases MINA53, NO66, EcYcfD, and TYW5 form homodimers through helix-rich motifs on the C terminus of their DSBH core[18,19] (structure PDB IDs: 1MZE, 4BXF, 4CCJ, 4CCL, and 3AL5, respectively). For ribosomal oxygenase TPA1P, the N-terminal domain is covalently linked to the catalytically inactive C-terminal domain, and the β strands of their β-helix cores stack to form an integrated unit, which in turn forms a homodimer through interactions between C-terminal domains[20] (structure PDB ID 4NHK).

L230 dimerization forms hydrophobic contacts between a leucine (L873) and an aromatic pocket formed by three tyrosines (Y798, Y801, and Y831) on the opposite subunit (Fig. 2a, inset, c). With a buried surface area of 268 Å², the L873 residues lock into the aromatic pockets on adjacent subunits. In LH family members, L873 and Y801 are conserved, but Y798 and Y831 are replaced with phenylalanine or tryptophan (Fig. 2d and Supplementary Fig. 3), preserving the aromatic pocket. L873D mutation caused loss of L230 dimerization (Fig. 2e and Supplementary Table 4) and enzymatic activity on collagen (Fig. 2f) but retained wild-type activity on a synthetic collagen

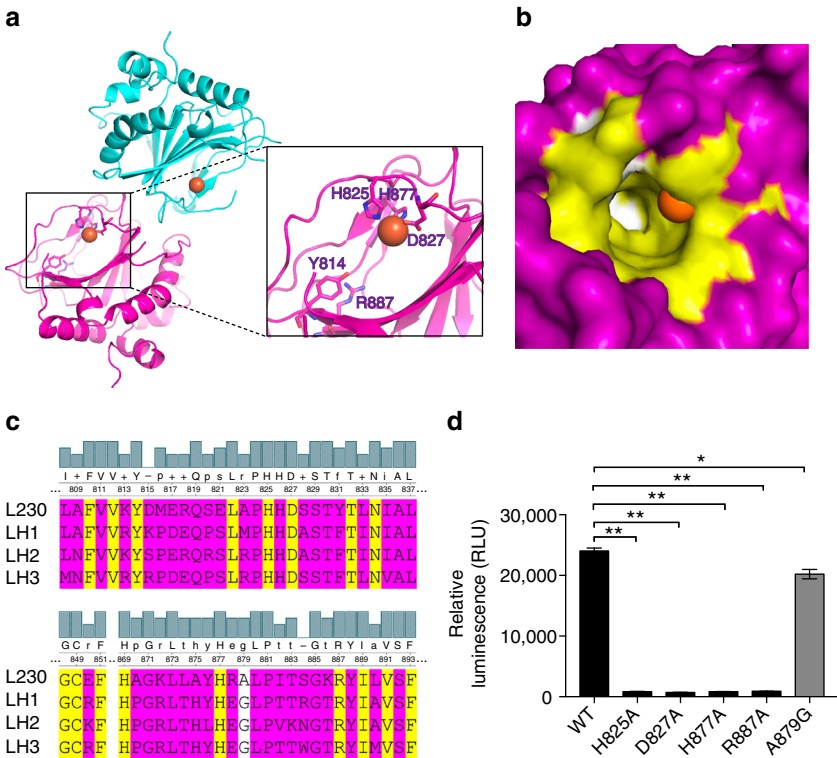

**Fig. 1** The active site of the L230 LH domain. **a** Ribbon diagram of the L230 LH domain. H825, D827, and H877 coordinate with $Fe^{2+}$ (orange); Y814 and R887 are positioned to bind to 2-OG (inset). **b** Surface view of L230 active site. Locations of $Fe^{2+}$ (orange) and the residues facing the 2-OG-binding pocket that are conserved (yellow) or not conserved (white) in human LH2 are indicated. Residues outside of 2-OG-binding pocket are in purple. **c** Sequence alignment of L230 with human LHs. Conserved (yellow) and non-conserved (A879, white) residues within the 2-OG-binding pocket and residues outside the 2-OG-binding pocket (purple) are indicated. **d** L230 enzymatic assay using bovine skin collagen as substrate. Enzymatic activity is ablated by loss of residues involved in $Fe^{2+}$- and 2-OG-binding but not a non-conserved residue (A879). Results are mean values ($\pm$S.D.) from biological triplicate samples. $p$ values, two-tailed Student's $t$ test

peptide (Fig. 2g), suggesting that the mutant protein's integrity was intact and that dimerization increases binding of lengthy collagen chains, which leads to increased apparent activity. Similarly, human LH2 dimer assemblies were lost following mutation of the conserved leucine (Supplementary Fig. 7 and Supplementary Table 4). Thus, LH proteins form homodimers through hydrophobic contacts, and dimerization enables LH activity on a natural collagen substrate.

**$Fe^{2+}$-binding stabilizes dimers to enhance collagen-binding**. A deep U-shaped surface cleft created by the dimerization was flanked by the active sites of each monomer (Fig. 3a), a unique architecture compared to other 2-OG-dependent oxygenase structures[16, 17]. Surprisingly, dimerization was lost following mutation of the $Fe^{2+}$-binding ligands (Fig. 3b and Supplementary Table 4) or $Fe^{2+}$ removal (Fig. 3c, Supplementary Fig. 8, and Supplementary Table 4), and dimerization was reconstituted by adding back $Fe^{2+}$ (Fig. 3c and Supplementary Table 4), whereas mutation of the predicted 2-OG-binding ligand did not lead to loss of dimerization (Fig. 3b and Supplementary Table 4), suggesting that $Fe^{2+}$-binding stabilizes the dimer. Similarly, LH2 dimerization was lost following mutation of a conserved aspartate (D689) in the amino acid triad (Supplementary Fig. 7). To exclude the possibility that $Fe^{2+}$ loss led to protein misfolding, we performed circular dichroism spectrometry and found no evidence to support that conclusion (Supplementary Fig. 9). We therefore reasoned that dimerization creates a cleft adjacent to the two active sites that may be a substrate-binding site. To test this,

we made L230 mutations that inhibited dimerization (L873D) or caused charge reversal in the cleft (K804E). Although achievable collagen concentrations did not saturate L230 enzymatic activity, the estimated $K_m$ of wild-type L230 ($2.6 \pm 0.6$ μM) was lower than that of K804E ($9.6 \pm 2$ μM) and L873D (Fig. 3d). The K804E cleft mutant reduced LH enzymatic activity on collagen (Fig. 3e) yet retained dimerization activity (Supplementary Fig. 10 and Supplementary Table 4) and enzymatic activity on the synthetic peptide (Fig. 3f), suggesting that the mutant protein's integrity was intact.

**L230 has telopeptidyl LH activity and promotes metastasis**. To determine whether L230 hydroxylates collagen telopeptidyl and/or helical Lys residues, we quantified collagen cross-links in orthotopic lung tumors generated by H1299 lung cancer cells that ectopically express full-length L230 or empty vector (Supplementary Fig. 11a). We analyzed the HLCCs dihydroxylysinorleucine (DHLNL), pyridinoline (Pyr), and deoxypyridinoline (d-Pyr); the LCC histidinohydroxymerodesmosine (HHMD); deH-dihydroxylysinonorlecine (HLNL), which can be an LCC or an HLCC depending on its derivation; and the precursor Hyl. Relative to controls, L230-transfected tumors had higher concentrations of Hyl (Fig. 4a), DHLNL, Pyr, and d-Pyr (Fig. 4b) but not HLNL (Fig. 4c), HHMD (Fig. 4d), or total collagen cross-links (Fig. 4e). Given that the HLCC-to-LCC ratio (Fig. 4f) and Hyl-to-collagen ratio (Fig. 4a) were higher in L230-transfected tumors than in controls, we conclude that L230 has both helical and telopeptidyl LH activities that result in a

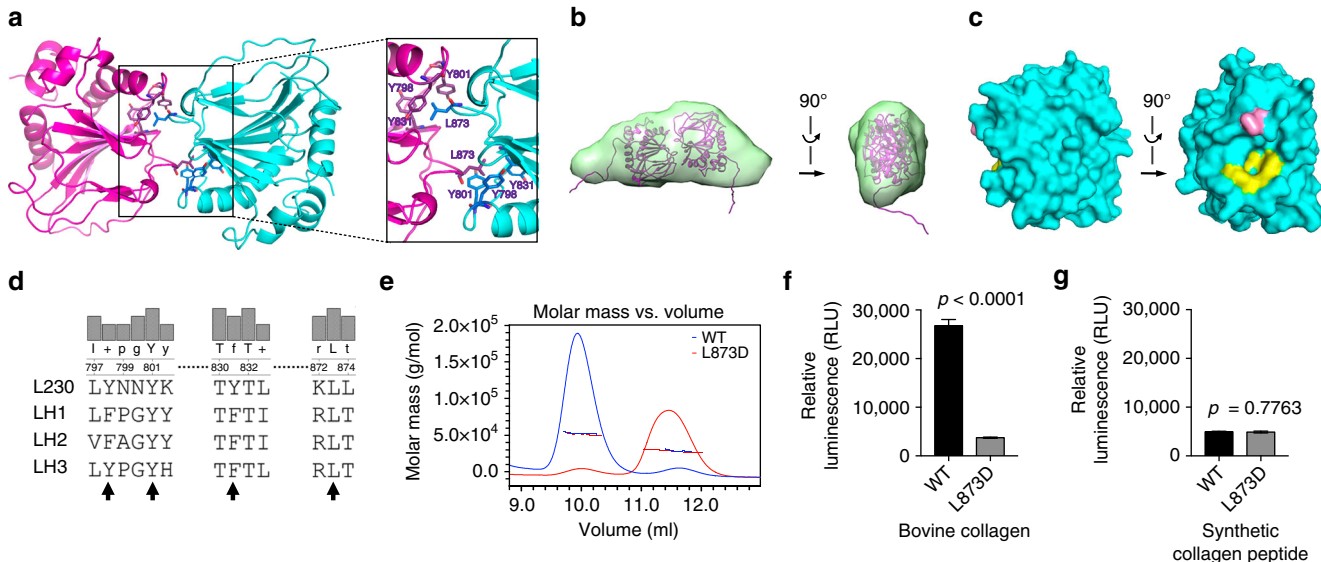

**Fig. 2** The LH domain forms a homodimer in a tail-to-tail orientation through a unique interface. **a** Ribbon diagram of the L230 LH domain homodimer. Key residues on the interface are indicated (inset). **b** A tail-to-tail homodimer detected by small-angle X-ray scattering (SAXS) analysis of L230 LH domain in solution. The modeled L230 dimer structure (magenta ribbon) is fit within the SAXS ab initio envelope (green). Two views rotated 90°. **c** Surface diagram of the L230 LH domain. Two views rotated 90°. Key tyrosine (yellow) and leucine (pink) residues on the interface are indicated. **d** Alignment of L230 with human LHs. L230 residues that mediate hydrophobic contacts on the dimerization interface are indicated (arrows). **e** Size exclusion chromatography with multi-angle static light scattering (SEC-MALS) of L230 LH domain. On the basis of elution time (X-axis) and molar mass (Y-axis), wild-type L230 (blue) forms a dimer, whereas L230 L873D (red) is monomeric. **f** Enzymatic assay of L230 LH domain proteins using bovine skin collagen as substrate. L873D mutation leads to loss of activity. Results are mean values (±S.D.) from triplicate biological samples. **g** Enzymatic assay of L230 LH domain proteins using synthetic collagen peptide as substrate. Retention of activity in the mutant suggests that protein integrity is intact. Results are mean values (±S.D.) from triplicate biological samples. *p* values, two-tailed Student's *t* test

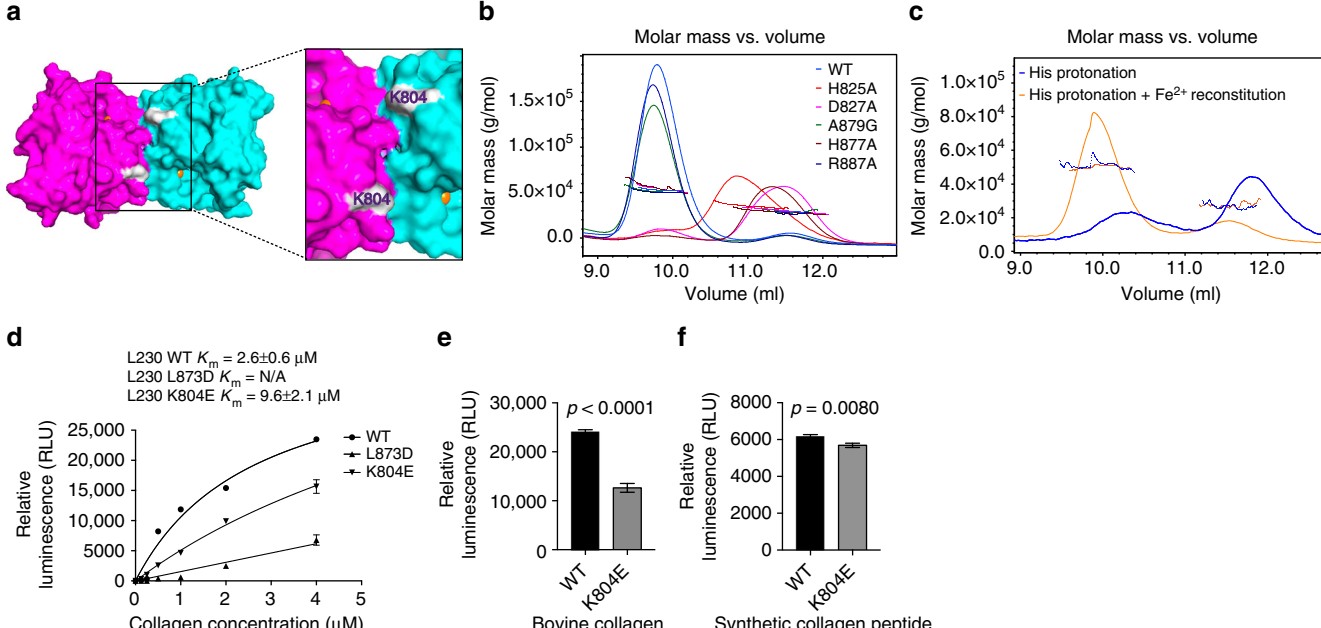

**Fig. 3** A cleft on the protein surface that may be involved in collagen binding. **a** Surface diagram of the LH domain homodimer. A cleft created by the dimerization interface is flanked by the active sites (inset). Locations of $Fe^{2+}$ atoms (orange) within each monomer (purple and cyan) are indicated. Location of K804 within the cleft (white). **b** SEC-MALS of L230 LH domain. On the basis of elution time (X-axis) and molar mass (Y-axis), dimerization is lost following mutation of residues in the amino acid triad (H825, D827, or H877) but not the residue positioned to bind 2-OG (R887) or non-conserved A879. **c** SEC-MALS of L230 LH domain after $Fe^{2+}$ removal. Dimerization is lost following histidine protonation to release the $Fe^{2+}$ (blue line) and restored following $Fe^{2+}$ reconstitution (orange line). **d** Enzymatic activity assays of L230 LH domain proteins using different concentrations of bovine skin collagen as substrate. Curves were used to calculate the $K_m$ values for wild-type L230 (WT) and L230 containing mutations on the dimerization interface (L873D) or within the cleft (K804E). **e**, **f** Enzymatic activity assays of L230 LH domain proteins using bovine skin collagen (**e**) or synthetic collagen peptide (**f**) as substrate. Relative to wild-type L230 (WT), K804E exhibited a reduction in activity on collagen, whereas activity in the context of a synthetic peptide was relatively preserved. Results in **b**–**f** are mean values (±S.D.) from triplicate biological samples. *p* values, two-tailed Student's *t* test

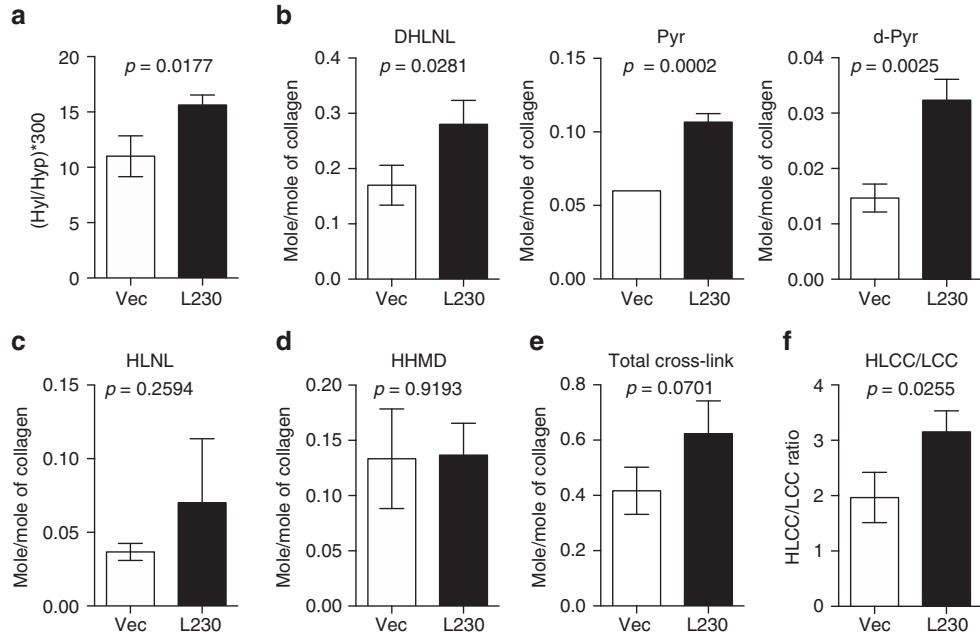

**Fig. 4** L230 induces a collagen cross-link switch in tumor stroma. Collagen cross-links (mol mol$^{-1}$ of collagen) in orthotopic lung tumors generated by injection of nude mice with H1299 cells stably transfected with empty vector (Vec) or full-length L230 (L230) (n = 3 tumors in each cohort). Quantification of **a** collagen Hyl content calculated as the product of 300 times the mean Hyl values divided by mean hydroxyproline (Hyp) values; **b** the HLCCs DHLNL, Pyr, and d-Pyr; **c** HLNL; and **d** the LCC HHMD. **e** Total collagen cross-links represent the sum of cross-links in **b**–**d**. **f** HLCC-to-LCC ratio was calculated as the sum of HLCC mean values (DHLNL + Pyr + d-Pyr) divided by LCC (HHMD) mean values. HLNL was excluded from the analysis because it can be classified as LCC or HLCC depending on its derivation. Results are mean values (±S.D.) from triplicate biological samples. p values, two-tailed Student's t test

collagen cross-link switch in tumor stroma, which recapitulates LH2 properties[10].

High LH2 levels correlate with reduced survival in multiple cancer types[7, 8, 10, 21], and LH2 promotes metastasis in murine tumor models[10]. Thus, we asked whether ectopic L230 expression enhances tumor metastatic properties. Compared to control transfectants, H1299 cells that ectopically express L230 generated orthotopic lung tumors with higher metastatic activity to the contralateral lung (Fig. 5a). Furthermore, they demonstrated greater migratory and invasive activities in Boyden chambers (Fig. 5b, c) but not colony formation in soft agar (Fig. 5d) or proliferation in monolayer (Fig. 5e). The pro-migratory and -invasive effects of L230 were abrogated by mutations in key Fe$^{2+}$- and 2-OG-binding ligands (Fig. 5b, c and Supplementary Fig. 11b), suggesting that LH activity is essential.

## Discussion

These findings provide insight into the structural properties of a collagen LH. They support a testable model in which dimerization stabilized by Fe$^{2+}$ incorporation creates a substrate-binding groove between subunit active sites. Given that the active sites are oriented in an anti-parallel mode, they could be loaded independently by two unzipped collagen chains or by looping of a single collagen chain. As the hydrophobic dimer contacts are conserved in LHs but no other Fe$^{2+}$- and 2-OG-dependent enzymes[17], we conclude that LHs evolved a dimerization mode that adapts to the unique properties of collagen. Whether other Fe$^{2+}$- and 2-OG-dependent oxygenase dimer assemblies are stabilized similarly by Fe$^{2+}$-binding remains unclear. Importantly, a mutation in a Fe$^{2+}$-binding ligand in human LH2 (D689A) leads to loss of LH activity and reduced ability to enhance tumor cell migration compared to wild-type LH2[8, 22]. Given that substrate specificities differ between human LHs (e.g., helical versus

telopeptidyl LHs) and between members of the larger LH family (e.g., L230 versus other LHs)[23, 24], these findings provide a framework for examining the structural basis for functional differences between LH family members.

On the basis of homology modeling, sequences located between the glycosyltransferase and LH domains previously shown to be required for LH dimerization[25] are not located on the dimerization interface of L230. This conclusion is supported by evidence from LH protein cross-linking experiments[25]. How these sequences regulate dimerization remains unclear, but it is possible that they contain binding sites for the peptidyl prolyl isomerase FKBP65 or other factors that promote LH2 dimerization. Unlike full-length LH2, which requires FKBP65 to form dimers, the L230 catalytic domain formed dimers in the absence of FKBP65, raising the possibility that FKBP65 is necessary for dimerization of the full-length LHs but not isolated catalytic subunits.

APMV was originally isolated from amoebae growing in the water of a cooling tower in Bradford, England[26]. Since the discovery of APMV, ~100 new mimiviral strains and related giant viruses (e.g., *Marseilleviridae, Myoviridae, Herpesviridae, Iridoviridae, Tobamoviridae, Ascoviridae*, and *Asfarviridae*) have been isolated from amoebae[27]. The giant viruses have structural and genetic features comparable to those of bacteria and small eukaryotes, an observation that has challenged the definition and classification of viruses[27, 28]. Genes that are expressed in giant viruses but no other viruses encode proteins involved in nucleotide synthesis, amino acid metabolism, protein modification, lipid or polysaccharide metabolism, DNA repair, and protein folding[29, 30]. It is debatable whether giant viruses represent ancient life forms or acquired their genes through horizontal gene transfer[30–33]. Our findings raise the possibility that L230 co-evolved with human LHs.

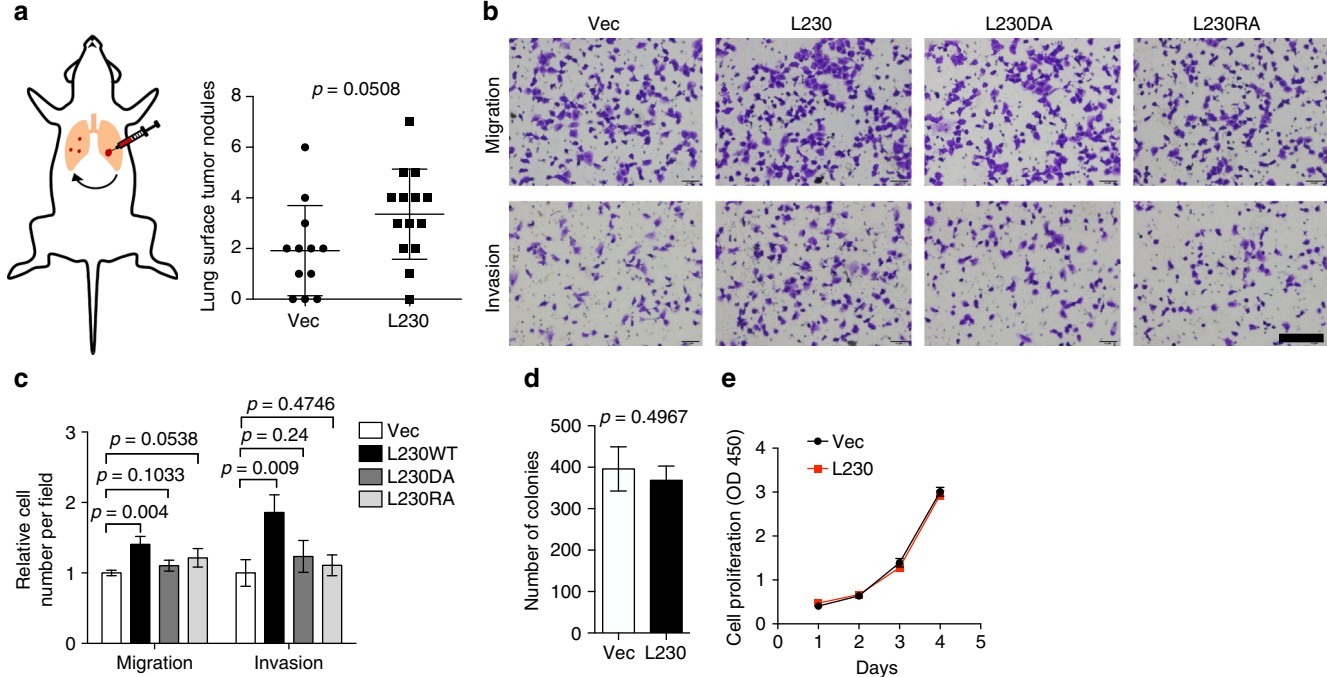

**Fig. 5** L230 promotes metastasis. **a** Scatter plot of numbers of lung metastases in each mouse (dots). Nude mice were injected intra-thoracically with H1299 cells that ectopically express full-length L230 or empty vector (Vec) to generate a single orthotopic lung tumor. After 4 weeks, mice were necropsied and visible tumors on the pleural surface of the contralateral lung were counted. **b, c** Migrated and invaded cells in Boyden chambers were imaged (**b**) and counted (**c**). H1299 cells that ectopically express full-length wild-type L230 (L230) or L230 containing mutations in an Fe²⁺-binding ligand (D827A, DA) or a 2-OG-binding ligand (R887A, RA) or empty vector (Vec) were seeded into the top chambers, and 10% fetal bovine serum was added to the bottom chambers as a chemoattractant. Scale bar = 400 μm. **d** Bar graph shows the average number of colonies formed by vector (Vec)- and L230-transfected H1299 cells in soft agar. **e** Relative densities of vector (Vec)- and L230-transfected H1299 cells in monolayer culture were determined at each time point by WST-1 assays. Results are mean values (±S.D.) from replicate biological samples. *p* values, two-tailed Student's *t* test

Pre-existing lung fibrosis from chronic obstructive pulmonary disease or idiopathic pulmonary fibrosis is a known risk factor for lung cancer development[34, 35], but infectious drivers of lung carcinogenesis have not been proven[36]. In serum samples, anti-mimivirus antibodies were originally detected in 9.6% of pneumonia patients and 2.3% of healthy controls[37]. Subsequent epidemiologic studies showed that mimivirus is uncommon in pneumonia[37, 38], but its prevalence in lung cancer is unclear. Collagen-like proteins and collagen-modifying enzymes have been detected in numerous pathogens, including mimivirus[15], and here we showed that mimiviral L230 has telopeptidyl LH activity, increases the formation of HLCCs, and promotes metastasis in an orthotopic lung tumor model. Analysis of human lung cancer genomic databases revealed no evidence of stable integration of L230 coding sequences, but the possibility of a transient APMV infection that could exacerbate intratumoral fibrosis without stably integrating mimiviral genomic elements merits investigation.

Therapies targeting enzymes that initiate collagen cross-linking (e.g., anti-LOX or -lysyl oxidase-like 2 antibodies) decrease tumor stiffness and suppress metastasis in mouse models of cancer[1, 6, 39, 40]. The structure identified here should aid in developing compounds that selectively inhibit human LHs for experimental biology and medicine.

## Methods

**Ethics statement**. All animal experiments were reviewed and approved by the Institutional Animal Care and Use Committee of the University of Texas MD Anderson Cancer Center. Mice were maintained and subjected to experiments using practices outlined by the National Institutes of Health at MD Anderson's Research Animal Facility. Mice bearing orthotopic lung tumors were monitored daily and killed at 4 weeks, which preceded signs of morbidity. Orthotopic lung tumors were not visualized or quantified prior to necropsy.

**Plasmids**. The full-length wild-type L230 was a gift (Thierry Hennet, University of Zurich). L230 was cloned with the Kozak consensus sequence (GCCACCATG) into pLVX vector modified to contain blasticidin resistance using SmaI and NotI cloning sites. A D827A mutation was introduced using PCR-based site-directed mutagenesis method. Truncated wild-type and D827A mutant L230 (aa680-895) was cloned into a modified pET-28b (Novagen) vector using XhoI and NotI cloning sites with standard PCR-based methods. This modified pET-28b vector has NheI inserted in the linker between His₆ and Thrombin recognition site, which changes the amino acid linker from GS to AS, and XhoI inserted in Thrombin recognition site without any changes in amino acid sequence. Other L230 mutant constructs were generated using QuickChange Lightning Site-Directed Mutagenesis Kit (Agilent). The identities of all constructs used in this study were confirmed by sequencing. Primers used for cloning and mutagenesis are listed (Supplementary Table 5).

**Protein expression and purification**. All of the truncated L230 proteins were overexpressed in *Escherichia coli* strain Rosetta-Gami 2 (DE3, Millipore). Cells expressing truncated L230 were induced with 1 mM isopropyl β-ᴅ-1-thiogalacto-pyranoside (IPTG) for 16 h at 16 °C. Cells were collected, pelleted, and then resuspended in binding buffer (20 mM Tris, pH 8.0, 200 mM NaCl, and 15 mM imidazole). The cells were lysed by sonication and then centrifuged at 23,000 × *g* for 15 min. The soluble proteins were first purified using Ni²⁺-resin (Qiagen) in binding buffer (20 mM Tris, pH 8.0, 200 mM NaCl, and 15 mM imidazole), and then further cleaned by gel filtration chromatography (Superdex 200, GE Health-care Life Sciences).

Human LH2 recombinant proteins were purified from Chinese hamster ovary cell-conditioned media as described previously with minor modifications[22]. In brief, LH2 recombinant proteins (residues 33–758, wild-type, D689A and L735D) were produced from new Gibco™ ExpiCHO™ cells in suspension (Thermo Fisher Scientific, Waltham, MA) as a secreted protein with N-terminal His₈ and human growth hormone (hGH) tags via large-scale transient transfection with polyethylenimine. The LH2-containing conditioned media were harvested by centrifugation at 7000 rpm for 10 min, filtered through 0.22 μm EMD Millipore

Stericup™ Sterile Vacuum Filter Units (EMD Millipore, Billerica, MA), concentrated to 100 mL, and buffer exchanged into nickel-binding buffer (20 mM Tris, 200 mM NaCl, 15 mM imidazole, pH 8.0) using the Centramate™ & Centramate PE Lab Tangential Flow System (Pall Life Sciences, Ann Arbor, MI). The recombinant LH2 proteins were then purified with immobilized metal affinity chromatography and gel filtration chromatography performed consecutively (Superdex 200, GE Healthcare Life Sciences) in a buffer containing 20 mM Tris, pH 8.0, and 200 mM NaCl.

**Crystallography.** Single high-quality crystals of the LH domain of L230 (amino acids 680–895) containing two protein molecules in the asymmetric unit were obtained via sitting drop vapor diffusion using a Mosquito crystallization robot (TTPLabtech) with a total 400 nl drop using a 1:1 mixture of L230 recombinant protein (15 mg ml$^{-1}$) with 0.2 M potassium citrate, 20% PEG3350 and 0.5% w v$^{-1}$N-dodecyl-N,N-dimethylamine-N-oxide. Native protein data were collected from flash frozen crystals on beamline 23-ID-B of GM/CA-CAT at the Advanced Photon Source, Argonne National Laboratory using an MX300 (Rayonix) CCD detector (Supplementary Table 6). For phase determination, crystals were grown in 0.2 M potassium citrate, 20% PEG3350, and 4.0% v v$^{-1}$ 2,2,2-trifluoroethanol and briefly soaked in crystallization solution plus 1.2 M of sodium iodine and then flash frozen. SAD data was collected at the 1.77 Å wavelength to enhance the phasing signal from the iodine with a Pilatus 6M (Dectris) pixel-array detector on the 23-ID-D beamline of GM/CA-CAT (Supplementary Table 6). Data were integrated and scaled using XDS. SHELX was used to locate iodine sites[41–43], obtain phase information, perform density modification, and generate an initial structural model, which was used as a search model for native structure. The structures were then built and refined via iterative model building and refinement using Coot[44] and PHENIX refine[45], respectively (Supplementary Table 6). L230 crystal structures shown are representative of biological replicates of native ($n = 10$) and iodine-soaked ($n = 3$) crystals.

**L230 enzymatic assay.** L230 enzymatic activity was measured as recently described[22]. In brief, the assay was performed in reaction buffer (50 mM HEPES buffer pH 7.4, 150 mM NaCl) at 37 °C for 1 h with 1 µM L230 enzyme, 10 µM FeSO4, 100 µM 2-OG, 500 µM ascorbate, 1 mM dithiothreitol, 0.01% Triton X-100, and 1 mM viral collagen peptide substrate ETGLKGII or 4 µM bovine skin collagen substrate containing no telopeptides (Bovine PureCol®, Advanced BioMatrix). With the exception of L230 recombinant protein and bovine skin collagen, all reagents were prepared immediately before use. All of these reagents were dissolved in reaction buffer with the exception of FeSO4 and collagen, which was prepared in 10 mM HCl, and the pH of the reaction mixture was checked with pH papers to ensure that HCl did not change the overall sample pH. Bovine skin collagen was denatured by heating at 95 °C for 5 min and then chilled immediately on ice before use. L230 activity was measured by detecting succinate production with an adenosine triphosphate-based luciferase assay (Succinate-Glo™ JmjC Demethylase/Hydroxylase Assay, Promega, Madison, WI) according to the manufacturers' instructions. Results shown are the mean values of triplicate biological samples in a single experiment. Each experiment was repeated once.

**SAXS analysis.** LH2 and L230 proteins were concentrated to 6 mg ml$^{-1}$ using Vivaspin 500 centrifugal concentrator with MWCO of 30 and 10 kDa, respectively. The concentrated protein was diluted with flow through buffer to prepare samples for three different concentrations. The flow through buffer was used as matching buffer for SAXS measurements and subtraction. SAXS data were collected at the SIBYLS beamline (12.3.1)[46] at the Advanced Light Source in Lawrence Berkeley National Laboratory. The sample was collected at 11 keV (1.127 Å) X-ray beam and the beam size at the sample was 4 mm × 1 mm converging to a 100-µm spot at the detector. The scattering vector $q$ range is from 0.01 to 0.5 Å$^{-1}$. The $q$ is defined as $(4\pi\sin\theta)\lambda^{-1}$, where $2\theta$ is the scattering angle and $\lambda$ is the wavelength. Each image collected was integrated and subtracted from buffer and inspected for radiation damage. The non-radiation damage data were merged using median for each concentration. SAXS data at three different concentrations were merged again using median. The final SAXS data were used for further analysis by ScÅtter (http://www.bioisis.net/tutorial/9) to extract $R_g$, $I(0)$, MW[47], and Porod volume and exponent[48]. The pair distribution plot $p(r)$ was converted using Gnom program[49] to estimate $D_{max}$ (maximum dimension), and the output Gnom file was used for ab initio shape modeling by Gasbor[50] and Dammif[51], followed by averaging ten models by DAMAVER[52]. For L230, the missing loop and N-terminal His-tag were modeled using MODELLER[53] implemented in chimera[54]. The best model fitted in FoXS[55] with better $\chi^2$ score was used to fit in ab initio SAXS envelope. Results are from triplicate biological samples in a single experiment. Analysis was performed once. The SAXS parameters and results are summarized in Supplementary Table 3.

**SEC-MALS analysis.** The molar mass of L230 and human LH2 proteins were analyzed by size exclusion chromatography coupled with multi-angle light scattering (SEC-MALS). Superdex 75 (10/300) column or Superdex 200 (10/300) was equilibrated with SEC-MALS buffer (20 mM Tris, pH 8, 200 mM NaCl, 0.02% sodium azide) at room temperature overnight using Akta purifier (GE Healthcare)

and connected downstream to MALS (DAWN HELEOS-II)-coupled to dynamic light scattering (DLS) detector and refractive index (RI) (Optilab T-rEX) instruments (Wyatt Technology) (Wyatt, 1993). L230 variants (3 mg ml$^{-1}$, 150 µl) were injected into the Superdex 75 SEC column with flow rate 0.5 ml min$^{-1}$ and the MALS data were analyzed by ASTRA 7.1 program using ribonuclease A (2 mg ml$^{-1}$, 13.7 kDa) as the isotropic scatter standard. Human LH2 proteins (2 mg ml$^{-1}$, 100 µl) were separated and analyzed similarly using a Superdex 200 SEC column. To chelate Fe$^{2+}$, L230 LH domain protein (3 mg ml$^{-1}$, 150 µl) was incubated with 25 mM EDTA and 50 mM dithiothreitol at room temperature for 16 h in an anaerobic glovebox. To release Fe$^{2+}$ via histidine protonation, L230 LH domain protein (0.3 mg ml$^{-1}$, 5 ml) was buffer exchanged into low-pH His-protonation buffer (10 mM sodium citrate, pH 4, 150 mM NaCl) and dialyzed for 16 h against 1 liter of low-pH His-protonation buffer as described[56] with minor modifications. After dialysis, L230 LH domain protein was buffer exchanged back into SEC-MALS buffer before running SEC-MALS. To reconstitute the Fe$^{2+}$-deficient L230 LH domain protein, protein was incubated with 300 µM ferrous ammonium sulfate and 50 mM dithiothreitol at room temperature for 2 h in an anaerobic glovebox. Results are from a single biological sample. Each protein was analyzed once. The calculated molecular masses and mass fractions (dimeric and monomeric) of LH2 and L230 proteins are summarized in Supplementary Table 4.

**Microscale thermophoresis.** Using techniques previously described[57], L230 protein was fluorescently labeled with Atto647 (Sigma) via amine coupling to achieve 1.4:1 labeling efficiency (protein-to-dye) as determined by NanoDrop (Thermo Scientific) absorbance at 280 and 647 nm. Labeled protein (40 nM, 10 µl) was mixed with equal volume of serially diluted (8.5 µM–0.5 nM) unlabeled L230 protein. After incubation at 25 °C for 15 min, the samples were loaded into silica capillaries (Nanotemper Technologies). Measurements were performed at 20 °C using Monolith NT.115 (Nanotemper Technologies). Data were analyzed (Nanotemper Analysis software. v.1.2.101) to fit $K_d$ according to the law of mass action. Results represent the mean values from triplicate biological samples in a single experiment. The experiment was repeated once.

**Circular dichroism.** L230 recombinant proteins were dialyzed into 0.01 M sodium phosphate and 150 mM NaCl (pH 7.4) and diluted to a concentration of 0.5 mg ml$^{-1}$. Circular dichroism spectra were measured using a J-810 spectropolarimeter (Jasco, Easton, MD) with a 2 mm path length quartz cuvette. All measurements were performed at 20 °C and three scans averaged for each spectrum. A blank spectrum of phosphate-buffered saline was collected in the same manner and used for background subtraction. Results represent the mean values from triplicate technical repeats in a single experiment. Each protein was analyzed once.

**Cell culture.** H1299 cells obtained from ATCC were authenticated by short-tandem repeat analysis and confirmed to be mycoplasma negative. Cells were grown in a humidified atmosphere with 5% CO$_2$ at 37 °C in Roswell Park Memorial Institute (RPMI) 1640 supplemented with 10% FBS. To establish stable transfectants, cells were infected with lentiviruses carrying cDNA that had been packaged in 293T cells using pMD2.G and psPAX2 plasmids. After 48 h, conditioned media were collected from the 293T cells and used for lentiviral infection of target cells. Stable transfectants were generated by selection with blasticidin (10 µg ml$^{-1}$) for 3–4 weeks.

**Orthotopic lung tumor model.** Immunodeficient (nu/nu) mice were randomized to balance the cohorts on the basis of age and gender. Mice were placed under general anesthesia (ketamine/xylazine 50 mg kg$^{-1}$ and 5 mg kg$^{-1}$, respectively, delivered by intraperitoneal injection) and an incision was made on the thorax under sterile conditions to expose the left lung[58]. Tumor cells (10$^6$) were injected directly into the left lung in 100 µl sterile PBS. The incision was closed using staples. The mice were humanely killed at 4 weeks after injection. Metastatic tumors visible on the surface of the right lung were manually counted. The primary tumor in the left lung was identified and carefully separated and removed from the lungs and flash frozen for collagen cross-link analysis. Investigators were blinded to the cohorts at the time of assessment of metastatic tumor numbers. The experiment was performed twice, and the results from the two experiments were combined to generate mean values from mice bearing L230-expressing tumors ($n = 14$) and control tumors ($n = 12$) tumors. Mice were excluded from the analysis if they died at the time of tumor cell injection due to hemorrhage or pneumothorax.

**Collagen cross-link and amino acid analyses.** Tissues were prepared, reduced with standardized NaB$^3$H$_4$, acid hydrolyzed and subjected to amino acid and cross-link analyses as reported[59]. The reducible cross-links, dehydro (deH)-dihydroxylysinonorleucine/its ketoamine, deH-hydroxylysinonorleucine/its ketoamine, and deH-histidinohydroxymerodesmosine (for cross-link chemistry, see ref. [60]) were analyzed as their reduced forms, i.e., DHLNL, HLNL, and HHMD, respectively, and the mature trivalent cross-links pyridinoline (Pyr) and deoxypyridinoline (d-Pyr) were simultaneously quantified by their fluorescence. All cross-links were quantified as mol mol$^{-1}$ of collagen based on the value of 300 residues of hydroxyproline (Hyp) per collagen molecule. The Hyl content in collagen was calculated

as Hyl Hyp$^{-1}$ × 300. Results represent the mean values from triplicate biological samples in a single experiment. Analysis was performed once.

**Proliferation assay**. Cells ($1 \times 10^3$) were seeded in 96-well plates ($1 \times 10^3$ cells per well) and incubated for defined time points; cell numbers were then measured using the WST-1 assay[58]. Briefly, cells ($2 \times 10^3$) were seeded into each well of a 96-well plate. WST-1 reagent (10 µl) (Roche) was added into each well, and the plates were incubated for 2 h. Viable cell density was assessed by determining the absorbance at 450 nm using a microplate reader (Bio-Rad). Results represent the mean values from triplicate biological samples in a single experiment. The experiment was repeated twice.

**Colony formation assay**. Cells in monolayer culture were harvested by brief digestion with 0.05% Trypsin-EDTA (Invitrogen), resuspended in medium containing 10% FBS, and aspirated through a 25G needle to generate single-cell suspensions[58]. These cells ($1 \times 10^4$) were mixed with 0.3% agar and plated in triplicate on 0.6% agar in medium supplemented with 20% FBS in 12-well plates. Cultures were maintained for 21 days to allow the formation of cell colonies, which were then visualized by crystal violet staining and manually counted under light microscopy. Results represent the mean values from triplicate biological samples in a single experiment. The experiment was repeated twice.

**Cell migration and invasion**. Cell migration and invasion assays were performed in Transwell and Matrigel-coated Boyden chambers, respectively, using 10% fetal bovine serum as a chemoattractant[58]. Transwell and Matrigel chambers (BD Biosciences) were used to quantify migratory and invasive activities, respectively. Cells ($2 \times 10^4$) were added to the upper chambers, and media containing 10% fetal bovine serum was added to the bottom chambers as a chemoattractant. After 8 h, cells that had reached the lower chamber were quantified by staining the porous membrane with 0.1% crystal violet, photographing the cells on the lower surface of the membrane, and counting the cells under a microscope. Results represent the mean values from triplicate biological samples in a single experiment. The experiment was repeated twice.

**Statistics**. Sample sizes were selected based on effect size and availability. Statistically significant differences between the mean values were determined by two-tailed Student's $t$ test. $p$ values <0.05 were considered significant.

**Data availability**. Crystal structures have been deposited in the Worldwide Protein Data Bank under RCSB accession numbers 6AX6 and 6AX7. All relevant data that support the findings of this study are available from the corresponding authors.

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

## Acknowledgements

This work was supported in part by the National Institutes of Health grants R21AR060978 (M.Y.), R01CA105155 (J.M.K. and M.Y.), P50CA70907 (J.M.K.), T32GM008280 (S.A.); Cancer Prevention and Research Institute of Texas (CPRIT) grant RP160652 (J.M.K.); The International Association for the Study of Lung Cancer Post-doctoral Fellowship (H.-F.G.); the Elza A. and Ina S. Freeman Professorship in Lung Cancer (J.M.K.); and The Welch Foundation F-1390 (K.N.D.); NIH GM123252 (K.N.D.); and CPRIT RP160657 (K.N.D.). We thank Kathryn Brunett and the SIBYLS beamline staff for collecting SAXS data. The SIBYLS beamline is supported through the Integrated Diffraction Analysis Technologies Program, which is supported by Department of Energy Office of Biological and Environmental Research, National Institute of Health project MINOS (R01GM105404), and a High-End Instrumentation Grant S10OD018483. We thank Andre Mueller from Wyatt Technology for MALS data analysis support.

## Author contributions

J.M.K. and H.-F.G. conceptualized the study. H.-F.G. designed, performed, and analyzed the L230 protein crystallography, amino acid sequence alignment, and L230 enzymatic activity assays. M.D.M., S.A., and G.N.P. analyzed the L230 protein crystallography. C.-L.T. and J.A.T. designed and performed the small-angle X-ray scattering assays. C.-L.T., H.-F.G., T.S.K., and K.N.D. designed and performed the dynamic light scattering experiments. C.-L.T. and H.-F.G. designed and performed the microscale thermophoresis experiments. M.T. and M.Y. designed, performed, and analyzed the collagen cross-linking assays. X.T., P.B., H.-F.G., X.L., J.Y., J.B., N.B.-R., Y.C., and J.M.K. performed and analyzed the tumor cell biology experiments.
