## [Peer Review File · Nature Communications]

Reviewers' comments:

Reviewer #1 (Remarks to the Author):

This is very nice paper, fully packed with novel information. For the very first time data are provided re the structure of lysyl hydroxylase. It provides not only insight into the structural properties of the active site, but also in the dimerization of the enzyme (which is required for its activity) and on one of the key residues involved in dimerization. Furthermore, it shows that a short form of lysyl hydroxylase produced by the Acanthamoeba virus is able to modulate the triple helix and the telopeptides of collagen in a mouse model when expressed ectopically. Furthermore, a set of experiments in vitro and in vivo shows proof that the lysyl hydroxylase activity of the enzyme is involved in metastatic activity and migratory activities.

A wealth information is provided. The manuscript is written in a clear way. The presented information will be of much help in developing ways to selectively inhibit lysyl hydroxylase activity so as to attenuate fibrosis as well as metastasis. I did not detect methodological flaws in the manuscript. The conclusions are justified based on the provided data.

Minor remark:

The name of *Drosophila melanogaster* should be changed into *Sophophora melanogaster* as of 2010 due to a decision of the International Commission on Zoological Nomenclature. The paper is enclosed.

[Editorial Note: International Commission on Zoological Nomenclature paper not reproduced in Peer Review File due to reprint policy restrictions.]

Reviewer #2 (Remarks to the Author):

Very interesting work on the lysyl-hydroxylase (LH) domain derived from the giant virus *Acanthamoeba polyphaga mimivirus*. Surprisingly, this virus contains a set of collagen-like molecules, which are lysyl-hydroxylated and further glycosylated by the same protein L230. The crystal structure of the LH domain of L230 is reported here. Although phylogenetically distant, the viral LH domain might serve as a model to invertebrate as well as vertebrate LHs as it has high level of conservation in secondary structure, and in Fe²⁺ and 2-oxoglutarate coordinating residues. Moreover, it might also represent a useful model for studying dimerization, as found in this study. A novel claim of this study is that dimerization is driven by the LH domain itself, opposed to what was reported previously for human and mouse LHs, where dimerization was attributed to a sequence between the glycosyltransferase activity and the lysyl hydroxylase activity domains. The authors have to clearly address this discrepancy. Two other levels of complications (not even mentioned in the manuscript!) are role of FKBP65 in driving dimerization of LH2, as well as FKBP65-dependent activity of LH2 (previous work of the authors). All these somewhat diminishes significance of L230 LH domain as a model for studying human LH2. Furthermore, activity and substrate specificity of viral L230 are quite different from LH2, which makes experiments with ectopic L230 expression quite limited.

Other major points:

1. As the authors, I also found very surprising that dimerization was lost following mutation of the Fe²⁺-binding ligands or Fe²⁺ removal, although Fe²⁺ is not involved in the dimerization interface. Nevertheless, Fe²⁺ binding might be important for formation of the tertiary structure. I found intriguing that at least one mutant H825A (one of Fe²⁺ coordinating residues) showed abnormal elution profile from gel-filtration column though with normal monomer mass as determined by light scattering. To better address role of Fe²⁺ in dimer formation (or even folding!) more experiments are required. These might include CD and fluorescence spectroscopy in order to assess secondary and

tertiary structures with and without Fe²⁺ (and wt vs mutants). Finally, role of Fe²⁺ in regulating pro-metastatic lysyl hydroxylase dimer assemblies, as claimed in the manuscript title, has also to be confirmed for LH2.

3. Based on SAXS derived protein envelopes the authors state that L230 LH domain forms tail-to-tail dimers similar to full-length human LH2. Very strong statement without any analysis and explanation.

4. L873D mutation leads to apparent decreased (but not loss of activity as stated in figure 2) activity for real collagen chain. Nevertheless, L873D activity becomes comparable to those for synthetic peptide, where the mutation has no effect. I would suggest to have the same scales for Y-axis in panels f and g (or combine the panels) to highlight it. This will support an idea that dimerization increases binding of "lengthy" or oligomerized collagen chains, which subsequently leads to increased "apparent activity".

5. An idea that substrate-binding groove positions collagen between subunit active sites is not obvious as active sites are located in an anti-parallel mode. Is there a possibility to load both sites in right orientation by forming a loop in a single collagen chain? Or maybe it is more biologically relevant to model binding of two adjacent chains (out of three chains forming a triple helix in a zipper fashion)?

Minor points:

1. Both structures were deposited with Fe³⁺ ions (FE (III) ION), should be Fe²⁺ (FE2 (II) ION).
2. Figure 1. The same colors (cyan and magenta) in panels a, b, and c represent different groups, very confusing.
3. Line 135. Not only in human!

Reviewer #3 (Remarks to the Author):

Guo et al have provided the first crystal structure information for an important family of collagen modifying enzymes in their manuscript. This work will influence the drug discovery efforts to target this enzyme which is known to play an important role in the metastasis of multiple cancer types. While the work is exciting and the crystal structure dataset is of relatively high quality based on the statistics presented in supplementary tables, there are several key points to address before publication.

1. In figure 1 the authors indicate the the Fe binding site and the 2-OG binding sites are both required for enzymatic activity. However, in Figure 3 they show that only the Fe binding site is required for dimerization. First, it would be useful if the authors could comment on the role of the 2-OG site since its clearly important for activity. It would also be appropriate to perform the collagen crosslink assays in figure 4 as well as the in vivo lung colonization assay in figure 5 with both the Fe and 2-OG binding site mutants . Doing so would clearly show that dimerization and the specific Fe and 2-OG binding sites that have been called out in this manuscript have functional importance.

2. The lung colonization study in Figure 5A specifically details that only tumors on the surface of the lung are quantified. This approach seems flawed. A more thorough investigation would require sectioning of the lungs and assessing tumor foci found throughout the organ. Additionally the authors could stain for expression of L230 to be sure that protein expression can verified in vivo. L230 expression should also be validated in the migration, invasion, colony forming, and proliferation studies included in this figure by Western blot, rather than PCR included in the supplemental information. However, that may be difficult due to the viral origin of the protein (see below).

3. Lastly, the authors use the viral LH (L230) for their biological validation studies in figure 5. It is unclear why they did not create the relevant mutants in the conserved sites of human LH1-3 and express those constructs in lung cancer cells to assign the necessary physiological relevance to their studies. Furthermore, it would be appropriate to silence LHs in these lung cancer cells to ask whether

they could reduce the number of metastatic foci #s from those found in the control cells. Overexpression assays are generally not thought reliable in isolation. The corresponding gene silencing assays would add depth and validity to those observations.

Reviewer #1

No critiques to address.

Reviewer #2

A novel claim of this study is that dimerization is driven by the LH domain itself, opposed to what was reported previously for human and mouse LHs, where dimerization was attributed to a sequence between the glycosyltransferase activity and the lysyl hydroxylase activity domains. The authors have to clearly address this discrepancy.

On the basis of homology modeling, sequences between the glycosyltransferase and LH domains previously shown to be required for LH dimerization (Heikkinen et al Matrix Biol. 30:27-33, 2011) are not located on the dimerization interface of L230. This conclusion is supported by evidence from LH protein cross-linking experiments (Heikkinen et al Matrix Biol. 30:27-33, 2011). How these sequences regulate dimerization remains unclear, but it is possible that they contain binding sites for the peptidyl prolyl isomerase FKBP65 or other factors that promote LH2 dimerization. Unlike full-length LH2, which requires FKBP65 to form dimers, the L230 catalytic domain formed dimers in the absence of FKBP65, raising the possibility that FKBP65 is necessary for dimerization of the full-length LHs but not isolated catalytic subunits. We have added a paragraph to the Discussion section to more fully address this point.

Two other levels of complications (not even mentioned in the manuscript!) are role of FKBP65 in driving dimerization of LH2, as well as FKBP65-dependent activity of LH2 (previous work of the authors). All these somewhat diminishes significance of L230 LH domain as a model for studying human LH2.

As discussed above, the reviewer has raised an important point. We have included a statement in the Discussion to address it.

Furthermore, activity and substrate specificity of viral L230 are quite different from LH2, which makes experiments with ectopic L230 expression quite limited.

The limited available comparative data do show that L230 and human LHs have distinct substrate specificities (Rutschmann et al. Appl Microbiol Biotechnol 98:4445-4455, 2014), which does limit the value of L230 as a model of human LHs. We have added a statement to the Discussion section to make this point.

As the authors, I also found very surprising that dimerization was lost following mutation of the Fe²⁺-binding ligands or Fe²⁺ removal, although Fe²⁺ is not involved in the dimerization interface. Nevertheless, Fe²⁺ binding might be important for formation of the tertiary structure. I found intriguing that at least one mutant H825A (one of Fe²⁺ coordinating residues) showed abnormal elution profile from gel-filtration column though with normal monomer mass as determined by light scattering. To better address role of Fe²⁺ in dimer formation (or even folding!) more experiments are required. These might include CD and fluorescence spectroscopy in order to assess secondary and tertiary structures with and without Fe²⁺ (and wt vs mutants).

We performed circular dichroism spectroscopy and found no evidence that L230 secondary structure was altered by mutations in the Fe²⁺-binding ligands or by histidine protonation to

deplete Fe^{2+} . We have added these findings to the Results section. Why the elution profile was abnormal for one Fe^{2+} -binding ligand (H825) but not the others is unclear. One possibility is that the mutation caused a minor change in conformation that resulted in a solvation change.

Finally, role of Fe^{2+} in regulating pro-metastatic lysyl hydroxylase dimer assemblies, as claimed in the manuscript title, has also to be confirmed for LH2.

We generated full-length recombinant human LH2 protein that contains a mutation (D689A) in the amino acid triad and showed that this mutation leads to loss of dimerization. We have added these findings to the Results section.

Based on SAXS derived protein envelopes the authors state that L230 LH domain forms tail-to-tail dimers similar to full-length human LH2. Very strong statement without any analysis and explanation.

We agree and now simply state that, on the basis of findings from SAXS analysis, L230 forms dimeric complexes.

L873D mutation leads to apparent decreased (but not loss of activity as stated in figure 2) activity for real collagen chain. Nevertheless, L873D activity becomes comparable to those for synthetic peptide, where the mutation has no effect. I would suggest to have the same scales for Y-axis in panels f and g (or combine the panels) to highlight it. This will support an idea that dimerization increases binding of “lengthy” or oligomerized collagen chains, which subsequently leads to increased “apparent activity”.

We appreciate these helpful suggestions and have modified the text and Figure 2 accordingly.

An idea that substrate-binding groove positions collagen between subunit active sites is not obvious as active sites are located in an anti-parallel mode. Is there a possibility to load both sites in right orientation by forming a loop in a single collagen chain? Or maybe it is more biologically relevant to model binding of two adjacent chains (out of three chains forming a triple helix in a zipper fashion)?

The reviewer's observation that the active sites are in anti-parallel mode does indeed raise the suggested possibilities. We have modified the text accordingly.

Both structures were deposited with Fe^{3+} ions (FE (III) ION), should be Fe^{2+} (FE2 (II) ION).

We appreciate the reviewer noting the typographical error in the structure deposits. This has been corrected.

Figure 1. The same colors (cyan and magenta) in panels a, b, and c represent different groups, very confusing.

Figure 1 has been modified so that distinct colors represent the different groups in panels a, b, and c.

Line 135. Not only in human!

The word “human” has been removed.

Reviewer #3

In figure 1 the authors indicate the the Fe binding site and the 2-OG binding sites are both required for enzymatic activity. However, in Figure 3 they show that only the Fe binding site is required for dimerization. First, it would be useful if the authors could comment on the role of the 2-OG site since its clearly important for activity.

We agree that our findings raise the possibility that 2-OG-binding ligands regulate dimerization. However, in Figure 3b, L230 dimerization was preserved following mutation of the 2-OG-binding ligand (R887). Thus, our findings suggest a specific role for Fe²⁺-binding ligands in dimerization.

It would also be appropriate to perform the collagen crosslink assays in figure 4 as well as the in vivo lung colonization assay in figure 5 with both the Fe and 2-OG binding site mutants . Doing so would clearly show that dimerization and the specific Fe and 2-OG binding sites that have been called out in this manuscript have functional importance.

To prove that the Fe²⁺- and 2-OG-binding site mutants, which are enzymatically inactive, lack biological activity, we stably transfected the Fe²⁺- and 2-OG-binding site L230 mutants into lung cancer cells and showed that they do not increase tumor cell migration or invasion, which are key pro-metastatic properties. These findings have been added to the Results section.

The lung colonization study in Figure 5A specifically details that only tumors on the surface of the lung are quantified. This approach seems flawed. A more thorough investigation would require sectioning of the lungs and assessing tumor foci found throughout the organ.

In the orthotopic lung tumor model reported here, tumor cells that metastasize to the contralateral lung lodge in alveolar capillary beds in peripheral lung tissues, generating metastatic deposits that are visible on the lung surface. We agree that counting tumors on the lung surface does not accurately assess *total* numbers of metastatic deposits in the contralateral lung. However, when used to assess experimental and control cohorts in the same manner, this approach does accurately reflect *relative* metastatic activity. We have reported this quantitative approach widely (Gibbons et al. *Genes Dev* 23:2140-2151, 2009; Yang et al. *J Clin Investig* 121:1371-1385, 2011; Ahn et al. *J Clin Investig* 122:3170-3183, 2012; Yang et al. *J Clin Investig* 124:2696-2708, 2014; Chen et al. *J Clin Investig* 125:1147-1162, 2015; Tan et al. *J Clin Investig* 127:117-131, 2017).

Additionally the authors could stain for expression of L230 to be sure that protein expression can verified in vivo. L230 expression should also be validated in the migration, invasion, colony forming, and proliferation studies included in this figure by Western blot, rather than PCR included in the supplemental information. However, that may be difficult due to the viral origin of the protein (see below).

Unfortunately, there are currently no anti-L230 antibodies available through commercial or proprietary sources.

Lastly, the authors use the viral LH (L230) for their biological validation studies in figure 5. It is unclear why they did not create the relevant mutants in the conserved sites of

human LH1-3 and express those constructs in lung cancer cells to assign the necessary physiological relevance to their studies.

We have already reported that a mutation in an Fe²⁺-binding ligand (D689) in human LH2 leads to loss of enzymatic activity (Guo et al Arch Biochem Biophys 618:45-51, 2017), and the Simon lab showed that the same LH2 mutant is unable to increase tumor cell migration (Eisenger-Mathason et al, Cancer Discovery 3:1190-1205, 2013). We have stated these points in the Discussion section.

Furthermore, it would be appropriate to silence LHs in these lung cancer cells to ask whether they could reduce the number of metastatic foci #s from those found in the control cells. Overexpression assays are generally not thought reliable in isolation. The corresponding gene silencing assays would add depth and validity to those observations.

We have already reported that short hairpin RNA-mediated silencing of LH2 in human and murine lung cancer cells leads to reduced tumor cell migratory, invasive, and metastatic activities (Chen et al. *J Clin Investig* 125:11471162, 2015).